# Design of 4.7 μm High-Efficiency Hybrid Dielectric Reflection Gratings

**DOI:** 10.3390/mi13040632

**Published:** 2022-04-16

**Authors:** Ye Wang, Xiuhua Fu, Yongyi Chen, Yuxin Lei, Li Qin, Lijun Wang

**Affiliations:** 1Changchun University of Science and Technology, Changchun 130022, China; wyoptics@163.com; 2Changchun Institute of Optics, Fine Mechanics and Physics, Chinese Academy of Sciences, Changchun 130033, China; leiyuxin@ciomp.ac.cn (Y.L.); qinl@ciomp.ac.cn (L.Q.); 3Peng Cheng Laboratory, No. 2, Xingke 1st Street, Nanshan, Shenzhen 518000, China; 4Academician Team Innovation Center of Hainan Province, Key Laboratory of Laser Technology and Optoelectronic Functional Materials of Hainan Province, School of Physics and Electronic Engineering of Hainan Normal University, Haikou 570206, China; 5Jlight Semiconductor Technology Co., Ltd., No. 1588, Changde Road, ETDZ, Changchun 130102, China

**Keywords:** diffraction grating, multilayer dielectric grating, high diffraction efficiency

## Abstract

Traditional reflective diffraction gratings working at 4.7 μm are fabricated by metal coatings. Due to the absorption of the metal itself, the diffraction efficiency (DE) could not reach over 95%. In this paper, we propose a 3 μm period multilayer grating design using hybrid multilayer dielectrics. With a layer of 0.353 μm Si and a layer of 0.905 μm SiO_2_ forming the rectangular grating, the maximum of larger than 99.99% and the overall first-order DE reached 97.88%. The usable spectrum width is larger than 0.2 μm, more than four times larger than that of the pure Si rectangular grating. This high DE multilayer grating is an ideal element for high-power laser systems with the spectrum beam combining method.

## 1. Introduction

High-power mid-infrared laser systems working at 4.7 μm are needed in spectral beam combining systems, spectral detection systems and wavelength modulation systems [1]. The diffraction grating is the key element for spectrum beam combining in order to realize the high-power high beam quality laser system [2]. The diffraction efficiency (DE) of the grating in the high-power spectral beam combining system directly affects the usable energy of the entire system [3]. Since the laser beams pass through the diffraction grating back and forth, the reduction of the diffraction efficiency will greatly reduce the output energy of the high-power system [4]. Thus, high DE will not only increase the laser system’s optical-electrical efficiency but also increase the ultimate output power. Thus far, the diffraction gratings working in this wavelength generally use Au-coating [5]. Due to the inherent absorption characteristics of the metal layer, not only the highest DE is limited to lower than 95%, but also the final output power is limited due to the low laser damage threshold caused by the heat which the metal absorbed [6]. 

On the other hand, multilayer diffractive gratings (MDG) [7] have a higher peak DE of over 99%, as well as higher laser damage threshold due to none material absorption [8]. In recent years, many researchers have carried out design research on dielectric gratings with high diffraction efficiency, as shown in Table 1. MDG is widely used in lenses [9], biosensors [10], metamaterials [11] and filters [12]. Therefore, MDGs have attracted increasingly more attention in recent years. Yet, most MDGs have limited working wavelength ranges. The high DE for over 96% is always limited for less than a few tens of nanometers in the spectrum [13]. This is inconvenient for usage if more laser elements are to be added in spectrum beam combining system to further increase the output power. 

Besides, there are very few reports about gratings working at the wavelength of 4.7 μm. To our best knowledge, there is no report about the MDG at this wavelength. 

Therefore, this paper proposes a design of MDG with high DE as well as wide working spectrum range. We used a hybrid grating material in order to reach both high DE and wide working spectrum range. The highest DE reached for over 99.99%. The wide working spectrum range is simulated: from 4.6 μm to 4.8 μm, the overall first-order DE reached larger than 96%. We analyze different grating parameters in detail, including incident angle, filling factor and surface grating groove etching depth. This work not only can be used to instruct the grating fabrication experiments in near further, but also offers a new way to design new types of gratings with both high DE and wide working spectrum range. 

## 2. Model Structure

MDG design is carried out according to a normal high DE grating structure [17]. The 2D simulation was carried out using COMSOL Multiphysics. The mesh was set with a maximum size of less than 1/6th of the wavelength in the according material. The incident beam along the x-axis was set to be the port boundary with periodic conditions. The substrate was set to be the perfect matched layers. The periodic conditions were set at both of the y directions. For high DE reflection gratings, usually, there are Distributed Bragg Reflector (DBR) layers, phase match layers and the grating structure. In order to design a realistic grating that can be fabricated, the grating grooves cannot be too deep. The grating depth should be easily controlled too. For this consideration, an etch stop layer is added into the MDG, between the grating and the phase match layer. Meanwhile, we limit the etching depth so that only grating groove width depth ratios smaller than 1:1 are counted. 

The material for each layer is then selected. At the wavelength of 4.7 μm, there are very few film materials that could be used. 

The substrate is selected to be JGS1, Si or CaF_2_. We take Si for the modeling. The DBR composed of the high refractive index film material (selected as Si, refractive index 3.4699 [18]) and the low refractive index film material (selected as SiO_2_, refractive index 1.3576 [19]). The DBR film system design composed of these two materials is superior to other materials in the total film thickness and the difficulty of process preparation. 

The etch stop layer is selected as HfO_2_ film with a refractive index of 1.8600. The grating is composed of silicon-based material (Si and SiO_2_) for practical processing procedure, which could be easily etched by the same Inductive Coupled Plasma (ICP) etching recipe. These two layers of materials form an equivalent material with adjustable refractive index, which gives additional dimension to adjust both the DE and the working spectrum range. The total structure is demonstrated as shown in Figure 1. 

Considering the limitation of i-line lithography and the high chromatic dispersion required, a period of 3 μm is fixed, which is about 333.33 lines per centimeter. At a wavelength of 4.7 μm, according to grating diffraction equation:$$n\wedge \left({\sin\theta }_{\mathrm{Diff}}\pm {\sin\theta }_{0}\right)=m\lambda$$
and when $\theta_{\mathrm{Diff}}$ = $\theta_{0}$ = $\theta_{\mathrm{Littrow}}$:$$2n\wedge {\sin\theta }_{\mathrm{Littrow}}=m\lambda$$

The Littrow diffraction angle θ is calculated to be 51.567°.

Then, we fix the etch stop layer of HfO_2_ at 0.05 μm. 

The polarization is also fixed: only the TE polarization is considered, where the electric field is parallel to the grating grooves. The analysis is carried out using the parameters in Table 2.

Having these materials and parameters fixed, the design went through the steps detailed in the following section.

## 3. Simulation Results and Discussion

### 3.1. Multilayer Dielectric High-Reflection DBR on DE

The first purpose of the reflective grating in this paper is to design a suitable DBR. Multilayer dielectric high-reflection films with a center wavelength of 4.7 μm on the surface of the substrate using high and low refractive index dielectric materials with good refractive index matching are designed to deposit alternately. For 100% reflectivity, theoretically, the larger the coefficient of the reflective film stack, the higher the obtained reflectivity. However, as the number of film layers increases, the accumulated stress inside the film stack increases, and the device is more prone to physical damage, resulting in unstable device usage. The film system design parameter requirements of this high-reflection film are shown in Table 3.

Different optical properties of materials, different application scopes, and the process involved are also crucial, so for dielectric thin film materials, they should have the characteristics shown in Table 4.

The derivation of the admittance of single-layer films is extended to the case of arbitrary multilayer films using the matrix method. After continuous linear transformation, the matrix expression is obtained as:(1)$$\left[\begin{array}{c} k\times E_{0}\\ H_{0} \end{array}\right]=\left\{\overset{K}{\underset{j=1}{\prod }}\left[\begin{array}{cc} {\cos\delta }_{j} & \frac{i}{\eta_{j}}{\sin\delta }_{j}\\ {i\eta }_{j}{\sin\delta }_{j} & {\cos\delta }_{j} \end{array}\right]\right\}\left[\begin{array}{c} k\times E_{k+1}\\ H_{k+1} \end{array}\right]$$
where K = 7. The reflectance expression is:(2)$$R=\left(\frac{\eta_{0}B-C}{\eta_{0}B+C}\right){\left(\frac{\eta_{0}B-C}{\eta_{0}B+C}\right)}^{*}$$

From the optical admittance, the reflectance Formula (2) can be written as:
(3)$$R=\left[\frac{1-{\left(\frac{n_{H}}{n_{L}}\right)}^{2S}\left(\frac{n_{H}^{2}}{n_{g}}\right)}{1+{\left(\frac{n_{H}}{n_{L}}\right)}^{2S}\left(\frac{n_{H}^{2}}{n_{g}}\right)}\right]$$

From expression (3), it can be seen that the larger the ratio of high and low refractive index materials, or the more layers of thin films, the higher the reflectivity and the wider the reflection bandwidth. The reflection bandwidth is calculated in expression (4):(4)$$\Delta g=\frac{2}{\pi }\arcsin\left(\frac{n_{H}-n_{L}}{n_{L}+n_{L}}\right)$$

It can be seen from Figure 2 that the reflection bandwidth is sufficient to meet the design requirements.

OptiChar is used to fit the refractive index of high and low refractive index film materials in full spectrum, and in order to obtain the refractive index dispersion distribution relationship [20]. Optilayer is used to design the reflective film; according to the design requirements, the expression of the film stack is selected as the periodic film system (HL)^s^ H as the basic structure, and the central wavelength reflection characteristic is good, as shown in Figure 2. 

A large refractive index difference is used to highly reflect the light diffracted to the substrate direction. The thickness of each layer in the DBR is a regular quarter-wavelength thickness, that is, 0.353 μm of Si and 0.905 μm of SiO_2_. We use 10 pairs of Si/SiO_2_ DBR layers to set up the simulation and modeling, using the mode expansion method. The reflectivity of incident light with a wavelength of 4.7 μm reaches 99.9999%.

### 3.2. High Reflective Diffraction Grating Based on Pure Si and SiO_2_

Let us first consider pure Si gratings, as shown in Figure 3. Our first issue is to determine the phase match layer of SiO_2_, so that when DE reaches maximum at 4.7 μm, the grating layer of Si could be minimum for easy fabrication. Parameters used in this simulation is shown in Table 5. 

For pure silicon gratings, it is shown in Figure 4 that there are basically two sets of solutions within the width depth ratio of less than 1:1. For both sets, the highest DE red shifts with the increase of grating depth. 

At around 4.7 μm, both 0.36 μm and 1.220 μm of pure Si gratings satisfy the highest DE of more than 99.99%. At grating depth of 0.36 μm, the DE at 4.6 μm is 30.52%, the DE at 4.8 μm is 81.58%. A range of 4.687–4.727 μm has a DE of more than 96%. At a grating depth of 1.22 μm, the DE at 4.6 μm is 57.06% and the DE at 4.8 μm is 74.26%. A range of 4.672–4.726 μm has a DE of more than 96%. Deeper grating depth has larger working wavelength range, yet the etching condition becomes tougher. 

### 3.3. Using Hybrid Materials Instead of Pure Si for the Grating

For pure Si grating, a range of 0.04 μm has the DE of more than 96%. Yet, this range is not enough for high-power spectrum beam combining. In this case, we use the hybrid material, that is, a layer of Si and a layer of SiO_2_ to form grating, so that this hybrid material can adjust the effective refractive index of the grating to get a wide hide DE working spectrum range. Figure 5 demonstrates the DE versus the height of different SiO_2_ h_2_ from 0.3 μm to 0.5 μm when the high of Si h_1_ changes from 0.08 μm to 0.25 μm, at an intervening of 0.01 μm. We picked up the solutions that are with the minimum DE larger than 85%. It can be seen that there are always some solutions that meet the high DE requirement when h_1_ is altered, which means a comparatively high processing tolerance.

It is shown that, when h_1_ is fixed, the peak DE mainly red shifts when h_2_ increases. The peak DE can always reach over 99.9% when total height of SiO_2_ and Si are within 0.57~0.58 μm. If h_1_ is less than 0.14 μm, the increase of h_2_ decrease the minimum DE at 4.8 μm. This is because the overall material index for the hybrid material is small when the perfect phase of the grating matches the highest DE. The highest DE usually appears larger than 4.7 μm. When h1 is more than 0.15 μm, the increase of h_2_ decrease the minimum DE at 4.6 μm. This is because the overall material index for the hybrid material is larger when the perfect phase of the grating matches the highest DE. The highest DE usually appears smaller than 4.7 μm. 

From these simulations, we picked up some lines when h_1_ is selected at 140 nm ± 10 nm and h_2_ is selected at 440 nm ± 10 nm, as shown in Figure 6. It can be see that, when the height of SiO_2_ h_2_ is selected at 140 nm ± 10 nm and the height of Si h_1_ is selected at 440 nm ± 10 nm, the highest DE around 4.7 μm reaches 99.99%, and the overall DE from 4.6 μm to 4.8 μm can reach over 96%. Notice that this ± 10 nm fabrication error is large enough that it can be easily controlled in experiment. Especially for conditions that are not so close to the largest error, suppose the total height of SiO_2_ and Si are within 570~580 nm, the overall DE from 4.6 μm to 4.8 μm can reach over 97%. Moreover, when h_1_ = 140 nm and h_2_ = 440 nm, the overall DE from 4.6 μm to 4.8 μm can reach over 97.88%. In this condition, the working spectrum (minimum DE > 96%) for high-power beam combining reaches for over 200 nm, at least four times larger than that of a pure Si grating. This DE is also higher than that of a metal grating, and since there is no absorption loss, the high-power laser damage threshold would be better. Notice that the total etching depth for h_1_ + h_2_ ≤ 0.6 μm is still smaller than half of the groove width (1.5 μm). This total height can be easily etched by one procedure of ICP. 

### 3.4. Influence of Grating Filling Factor

For actual fabrication, there are inevitable fabrication errors on the grooves’ width, which mainly affect the filling factor, as well as the DE. The question is how much the impact can be, and whether this impact can be eliminated. We fixed the h_1_ = 140 nm and h_2_ = 440 nm. Simulation about different filling factors at different incident angles versus wavelengths are carried out as shown in Figure 7. 

It can be seen that, if the filling factor is smaller than 0.5, there is always a certain incident angle for the total spectrum to have a high DE for over 97%, and for 4.7 μm, there is always an appropriate incident for the DE to be over 98.5%. There is always a certain angle across the 4.6 μm to 4.8 μm to have the highest DE be larger than 99.9%. Yet, for the filling factor larger than 0.5, the smallest DE at 4.6 μm drops dramatically. For wavelengths smaller than 4.7 μm, it will be hard to have a DE larger than 98% again. These simulation shows that the changes of the diffraction angle give a tougher processing tolerance, which may be caused by overexposure or ICP etching.

## 4. Conclusions

In this paper, we numerically designed a high DE grating. By introducing a hybrid material of one layer of SiO_2_ and one layer of Si, the maximum DE reached over 99.99% and over all spectrum from 4.6 μm to 4.8 μm reached over 97.88%. This introduction of hybrid material multilayer grating instead of metal-coated grating or pure Si grating not only avoids the metal absorption, which further increases the DE and laser damage threshold, but also increases the usable working spectrum for at least five times. Our proposed grating design is suitable for high-power laser spectrum beam combining with a large processing tolerance. Future work will be focused on the grating fabrication, the experiment tests as well as the high-power laser spectrum beam combining systems. 

## Figures and Tables

**Figure 1 micromachines-13-00632-f001:**
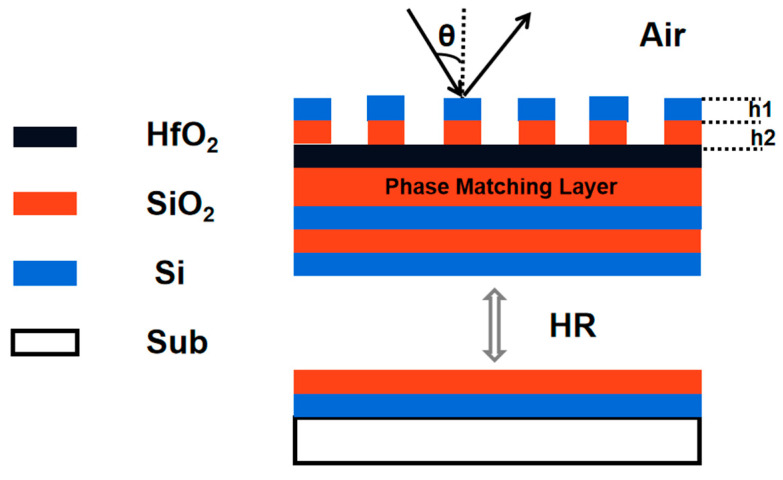
Brief structure of the MDG. The reflective DBRs are on the substrate, with a layer of phase match SiO_2_ and a layer of etch stop HfO_2_ subsequently. The grating consists a hybrid layer of Si and SiO_2_.

**Figure 2 micromachines-13-00632-f002:**
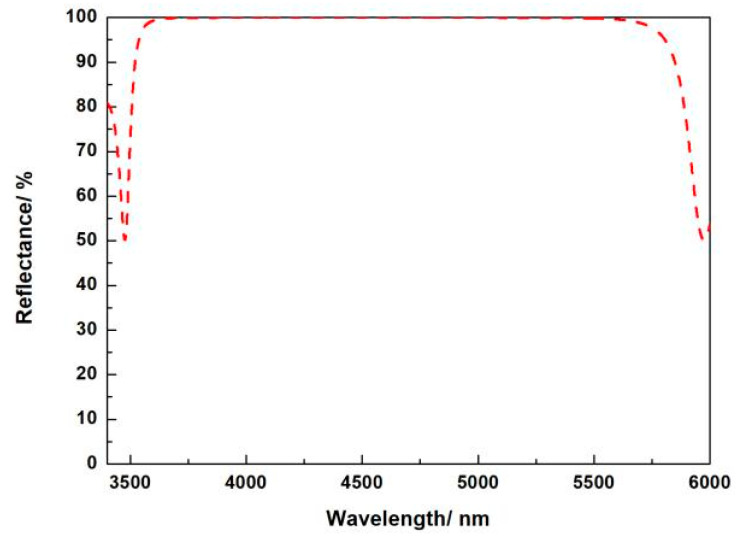
Reflectance spectral curve.

**Figure 3 micromachines-13-00632-f003:**
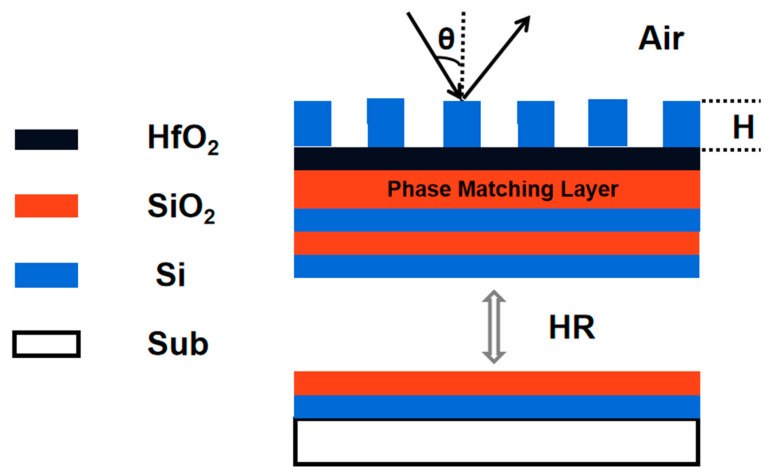
Brief structure demonstrating the pure silicon grating for comparation. The grating consists only of silicon material.

**Figure 4 micromachines-13-00632-f004:**
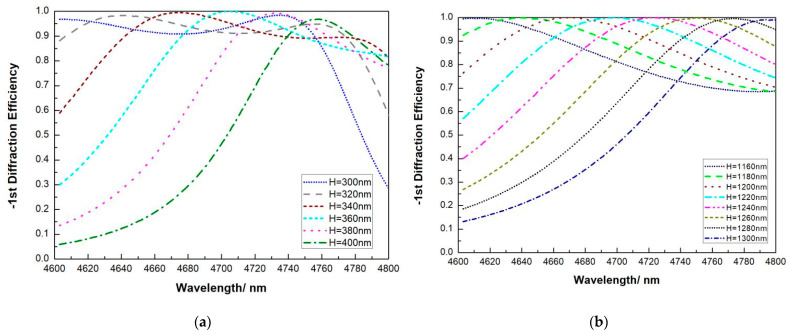
First-order DE versus wavelength with pure Si grating when the grating depth H changes (**a**) from 300 nm to 400 nm and (**b**) from 1160 nm to 1300 nm.

**Figure 5 micromachines-13-00632-f005:**
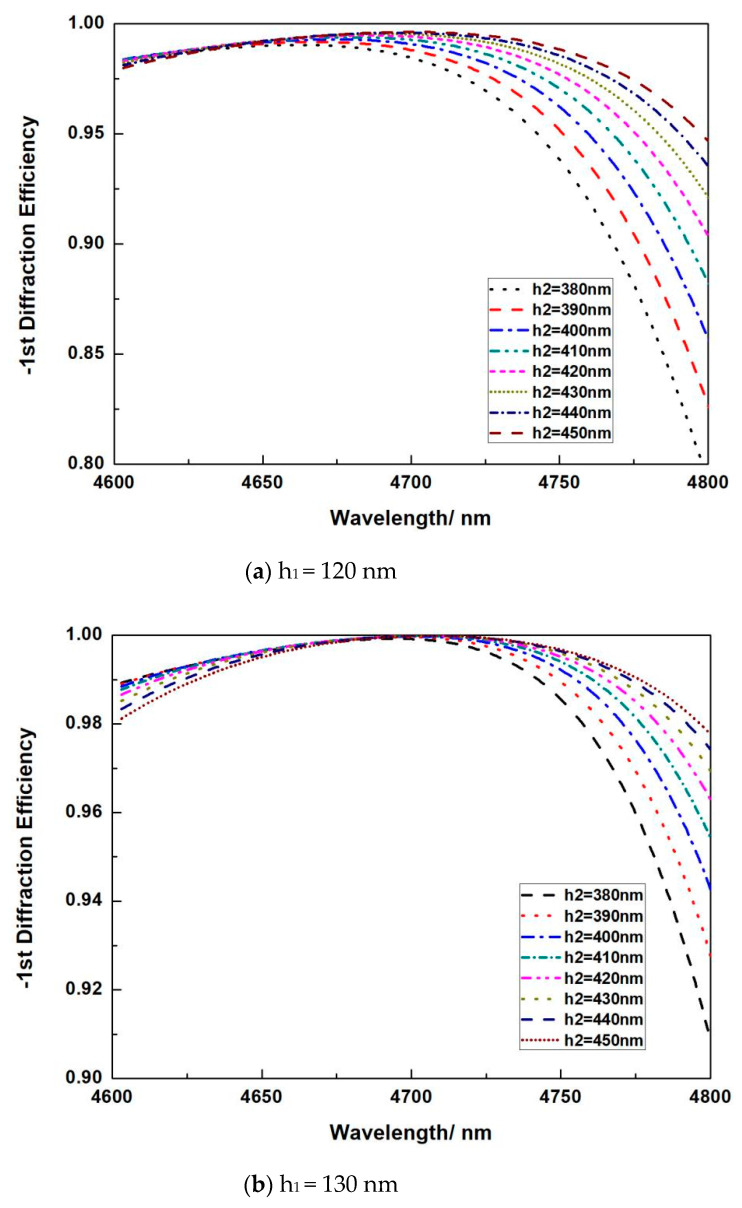
Relationship between the DE versus wavelength when height of SiO_2_ h_2_ changes with fixed Si h_1_ at (**a**) 120 nm, (**b**) 130 nm, (**c**) 140 nm, (**d**) 150 nm, (**e**) 160 nm, (**f**) 170 nm, (**g**) 180 nm and (**h**) 190 nm.

**Figure 6 micromachines-13-00632-f006:**
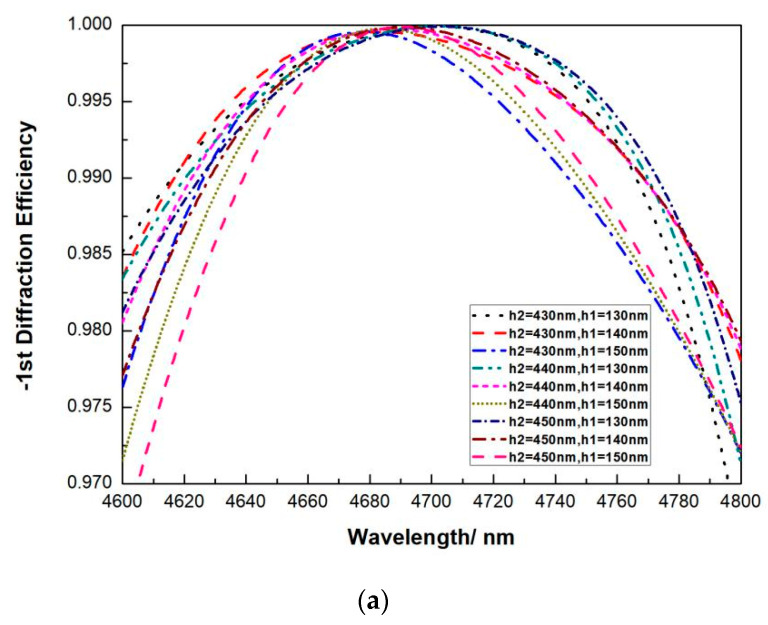
Relationship between the DE versus wavelength, at (**a**) h_1_ = 140 nm ± 10 nm and h_2_ = 440 nm ± 10 nm and (**b**) h_1_ = 140 nm and h_2_ = 440 nm. (**c**) Electric field distribution profile for four periods of the incident wavelength at 4.7 μm, 51.57° incident angle, at h_1_ = 140 nm and h_2_ = 440 nm.

**Figure 7 micromachines-13-00632-f007:**
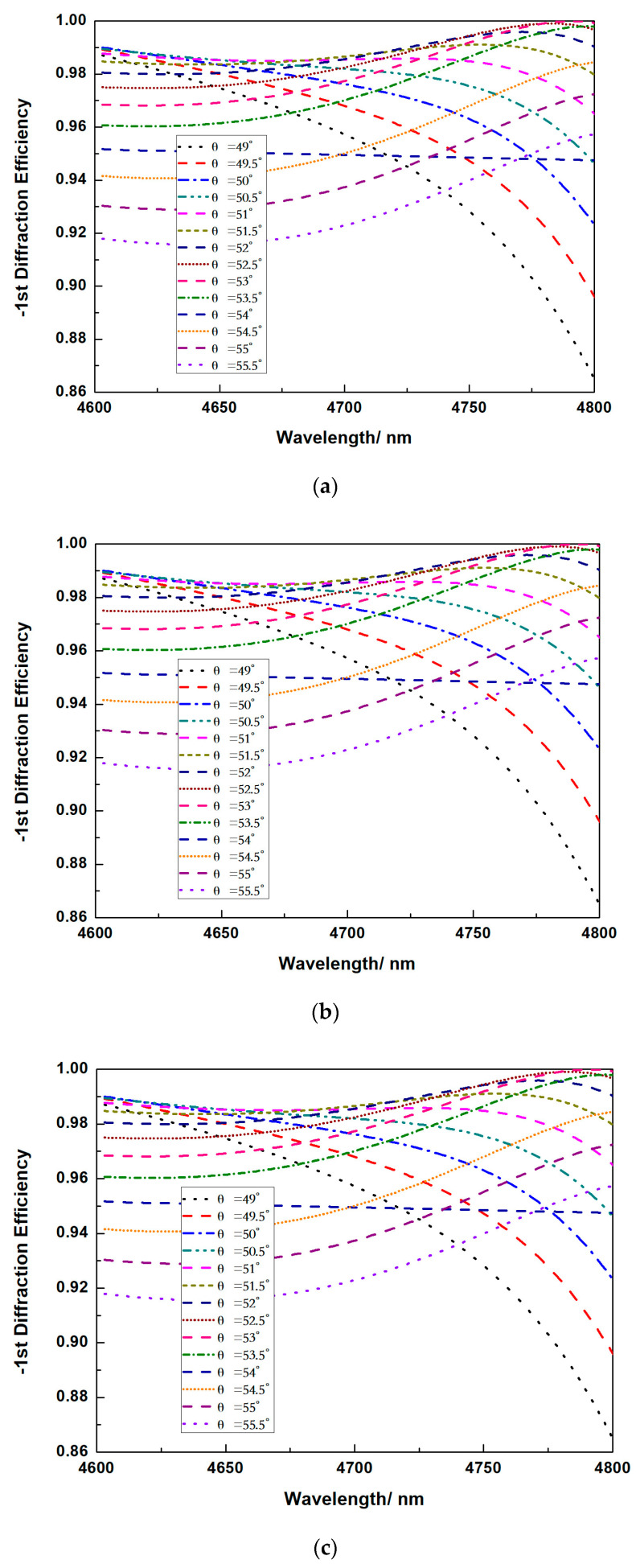
Relationship between the DE versus wavelength at (**a**) ff = 0.35, (**b**) ff = 0.4, (**c**) ff = 0.45, (**d**) ff = 0.5, (**e**) ff = 0.55, (**f**) ff = 0.6, (**g**) ff = 0.65.

**Table 1 micromachines-13-00632-t001:** Research on the design of dielectric gratings in recent years.

Researcher	Grating Design
Qunyu Bi, et al. [14]	RIRG etched into fused silica, first-order DE > 98% range over 170 nm (λ_0_ = 800 nm)
Changxi Xue, et al. [15]	First-order PIDE > 95% ZnSe/ZnS Microstrcture, 3–5 μm and 8–12 μm
V., et al. [16]	First-order DE > 97% 160 nm range (λ_0_ = 800 nm), 8 pairs of TiO_2_/SiO_2_ + TiO_2_

**Table 2 micromachines-13-00632-t002:** Parameter of reflective grating.

Spectral Range	4.6–4.8 μm
Incident angle	51.567°
Substrate	Si
Period	3 μm
Etch stop layer	0.05 μm

**Table 3 micromachines-13-00632-t003:** Parameter requirements for high-reflection film.

Spectral Range	4.6–4.8 μm
Reflectance	99.98%
Incident angle	51.567° in air
Substrate	Si

**Table 4 micromachines-13-00632-t004:** Requirements for the characteristics of dielectric film materials.

Characteristic	Requirment
Refrective index	Uniform, repeatable
Reflectance	High, k < 10^−4^
Scattering	Little, 10^−4^ for λ/4
Stress	Low
Adhesion	High
Anti-Laser Radiation Capability	As high as possible
structural defects	As little as possible

**Table 5 micromachines-13-00632-t005:** Parameter of reflective grating to determine the phase match layer of SiO_2_.

Spectral Range	4.6–4.8 μm
DBR pairs	10
Incident angle	51.567°
Substrate	Si
Period	3 μm
Filling Factor	0.5
Etch stop HfO_2_	0.05 μm
Phase match SiO_2_	2.2 μm
width depth ratio	<1:1

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
