# Peer review of "Design of 4.7 μm High-Efficiency Hybrid Dielectric Reflection Gratings"

_micromachines, 2022, doi:10.3390/mi13040632_

Round 1
Reviewer 1 Report
The authors present numerical study on a reflective diffraction grating operating a wavelength of 4.7 µm based on a periodic multilayer grating. Si and SiO2 layers form a rectangular grating. The absorption of the designed structure is minimum since no metal is used in the design. The proposed grating could be used in high power laser systems with spectrum beam combining method. The authors should address the following comments before publication:
- The captions of some figures are very brief. More information could be provided in the captions.
- 2 can be removed. A reference to the dispersion of Si and SiO2 is sufficient.
- What is the simulation software? And what are the simulation settings such as mesh, boundary conditions, 2D/3D, and etc.?
- I recommend to add the electric field distribution for the final design.
- Comparison with previous studies in a table is recommended which would help to highlight the advantages or disadvantages of the designed structure.
- The Introduction should be improved by mentioning previous studies. For instance:
- https://doi.org/10.1364/AO.58.002589
- https://doi.org/10.1109/JPHOT.2010.2059003
- https://doi.org/10.1364/AO.53.005769
- https://doi.org/10.1364/OL.44.003014
- https://doi.org/10.1364/OL.36.001431
- Mentioning other applications of grating structures- such as lenses, biosensors, metamaterials, and filters- would enrich the Introduction. For instance:
- https://doi.org/10.1364/OE.18.023529
- https://doi.org/10.1364/AO.434927
- https://doi.org/10.1109/JPROC.2018.2851614
- https://doi.org/10.1364/JOSAB.419475
- Some sentences require modification, such as:
- The structure of the reflective grating in this paper is firstly be designed.
- It can be seen that, (incomplete sentence)
Author Response
Dear reviewer,
Please see the attachment.
Warmest,
Ye Wang

Reviewer 2 Report
Reviewer’s comments on “Design of 4.7 μm high-efficiency hybrid dielectric reflection gratings” (Micromachines-1675781)
Summary:
In this manuscript, a numerical method is proposed to design a reflection grating with very high diffraction efficiency. By properly selecting and arranging the dielectric layer, the grating has very high diffraction efficiency at a wavelength of 4.7 μm. Now it is only the simulation result of the grating, and it is more expected that there will be experimental verification of the designed grating in the future.
The manuscript is well organized and easy to understand. But some typing errors that a copyeditor should be able to work with the manuscript. The paper is useful for the readers of Micromachines. However I have some comments for the authors:
- Please keep the same representation of numbers in terms of significant figures for all the results.
- Figures 2 and 3 are needed to be adjusted in order and corrected in the text as well.
- In Fig. 7, the range of incident angles used under different fill factors should be consistent.
- “So some kind of experimental error (such as over exposure, keeping the filling factor from 0.35~0.5) can be adjusted back when changing the diffraction angle” such a claim requires evidence to prove.
Author Response

(The authors gave the same response as above.)
